# Burnout among intensivists and critical care fellows in South Korea: Current status and associated factors

**Song I. Lee**[1‡], **Won-Young Kim**[2‡], **Duk ki Kim**[1], **Gee Young Suh**[3], **Jeongmin Kim**[4], **Ha Yeon Kim**[5], **Nak-Joon Choi**[6], **Won Kyoung Jhang**[7], **Sang-Hyun Kwak**[8], **Sang-Bum Hong**[9] *

1 Department of Pulmonary and Critical Care Medicine, Chungnam National University Hospital, Chungnam National University School of Medicine, Daejeon, Republic of Korea, 2 Department of Internal Medicine, Chung-Ang University Hospital, Chung-Ang University College of Medicine, Seoul, Republic of Korea, 3 Division of Pulmonary and Critical Care Medicine, Department of Medicine, Samsung Medical Center, Sungkyunkwan University School of Medicine, Seoul, Republic of Korea, 4 Department of Anesthesiology and Pain Medicine, Yonsei University College of Medicine, Seoul, Republic of Korea, 5 Department of Anesthesiology and Pain Medicine, Ajou University School of Medicine, Suwon, Republic of Korea, 6 Division of Acute Care Surgery, Department of Surgery, Korea University Guro Hospital, Seoul, Republic of Korea, 7 Department of Pediatrics, Asan Medical Center, University of Ulsan College of Medicine, Seoul, Republic of Korea, 8 Department of Anesthesiology and Pain Medicine Chonnam National University Medical School & Hospital, Gwangju, Republic of Korea, 9 Department of Pulmonary and Critical Care Medicine, Asan Medical Center, University of Ulsan College of Medicine, Seoul, Republic of Korea

‡ SIL and WYK contributed equally to the writing of the paper as first authors.
* sbhong@amc.seoul.kr

**Data Availability Statement:** The data analyzed in this study are the property of the Korean Society of Critical Care Medicine (KSCCM) and are not publicly available due to ethical and legal

## Abstract

Burnout among critical care physicians is an important issue that affects patient care and staff well-being. This study, conducted by the Korean Society of Critical Care Medicine, aimed to investigate the prevalence and associated factors of burnout among intensivists and critical care fellows in South Korea. From May to July 2019, a cross-sectional survey was conducted in 51 hospitals and 79 intensive care units offering subspecialty training in critical care medicine. Invitations were sent by email and text, and responses were collected using NownSurvey and Google Forms. Of the 502 invited participants, 253 responded (response rate: 50.4%). Significant contributing factors of burnout included being in an intensivist position (assistant professor/fellow) (odds ratio [OR], 3.916; 95% confidence interval [CI], 1.485–10.327; p = 0.006), working in a medical ICU (OR, 4.557; 95% CI, 1.745–11.900; p = 0.002), the number of stay-home night calls per month (OR, 1.070; 95% CI, 1.005–1.139; p = 0.034), and recent conflicts with colleagues (OR, 5.344; 95% CI, 1.140–25.051; p = 0.033). Similar factors were found to influence severe levels of burnout. This nationwide study indicates that a significant proportion of critical care physicians in South Korea experience burnout. Strategies to reduce overtime and workplace conflict are imperative to reduce burnout among these physicians and protect their mental health. Future research should explore targeted interventions for these specific factors.

restrictions. These restrictions are in place to protect potentially sensitive information and to comply with KSCCM's data sharing policy. Requests for data access may be directed to the KSCCM Data Access Committee at [email address: ksccm@ksccm.org]. Data access will be granted after review and approval by KSCCM and, if necessary, the relevant ethics committee.

**Funding:** This research was supported by the Korean Society of Critical Care Medicine (Grant No. KSCCM-2024-01). The funders had no role in study design, data collection and analysis, decision to publish, or preparation of the manuscript.

**Competing interests:** The authors have declared that no competing interests exist.

**Abbreviations:** ICU, intensive care unit; IRB, institutional review board; MBI, Maslach Burnout Index; EE, emotional exhaustion; DP, depersonalization; PA, personal accomplishment; OR, odds ratio; CI, confidence interval.

## Background

The medical profession, and particularly the intensive care unit (ICU), is naturally stressful, resulting in high rates of burnout among healthcare professionals [1, 2]. Burnout is a multidimensional syndrome characterized by emotional exhaustion (EE), depersonalization (DP), and a decreased sense of personal accomplishment (PA) [3]. It is associated with not only depression and absenteeism but also with increased turnover and early retirement [4, 5]. Additionally, it can lead to decreased professionalism [6], a higher incidence of medical errors [7], and a lack of compliance with safety standards [8].

Internationally, burnout rates among ICU professionals are alarmingly high, with studies reporting prevalence rates ranging from 33% to 47% in Western countries [9–11]. The COVID-19 pandemic has further exacerbated burnout, with one study [12] showing that more than half of nurses experienced emotional exhaustion and 85% experienced depersonalization. These findings highlight the critical need for targeted interventions for ICU staff. In Asia, a recent cross-sectional survey of 159 ICUs in 16 countries and regions reported a burnout prevalence of 50–52% among ICU physicians and nurses [13]. Despite these regional data, there is a lack of specific research focusing on South Korea. This gap is significant given the potential influence of cultural, economic, and systemic factors unique to South Korea. For example, cultural values such as collectivism and perseverance, which are prominent in South Korea, may promote social support but also increase vulnerability due to heightened expectations and self-imposed pressures [14, 15]. In addition, the diversity of health care systems and economic conditions across Asia limits the generalizability of findings from regional studies [16–18]. With its advanced healthcare system and unique workplace dynamics, South Korea warrants focused investigation to identify specific contributors to burnout and develop tailored strategies for prevention and intervention [19, 20].

Given the prevalence of burnout among medical professionals, this study focuses specifically on the unique challenges faced by intensivists and critical care fellows in South Korea. While previous studies have examined burnout among intensivists in other contexts, data specific to South Korea remain limited. The South Korean healthcare system is currently under significant pressure [21], with ongoing medical disputes adding to the stress and workload of critical care clinicians. This increased burden has contributed to staff turnover in the ICU, exacerbating the challenges faced by intensivists and fellows. By assessing the prevalence and associated factors of burnout in this population, this study aims to provide critical insights for the development of targeted interventions aimed at reducing burnout, improving the well-being of healthcare professionals, and improving patient care outcomes.

## Methods

### Study design

This study was based on a survey conducted by the Korean Society of Critical Care Medicine. The survey targeted hospitals nationwide that provide subspecialty training in critical care medicine, focusing on intensivists and critical care fellows actively working in ICUs. Data were collected from May to July 2019, in order to gain comprehensive insights into the current state of ICUs and the prevalence of burnout syndrome among ICU physicians. To encourage participation, the survey was announced on the Announcements section of the official website of the Korean Society of Critical Care Medicine (https://www.ksccm.org/), and additional efforts were made to engage participants by sending email and text message invitations directly to intensivists and critical care fellows. The survey responses were collected using NownSurvey (https://www.nownsurvey.com/), a proprietary online survey platform, and Google Forms, a

publicly available web-based tool for the creation and management of surveys. Of the 502 individuals invited to participate, including intensivists and critical care fellows, 253 responded, resulting in a response rate of 50.4%.

## Ethical considerations

This study was exempt from Institutional Review Board (IRB) review following submission of an exemption request to the Chungnam National University Hospital IRB (No. 2023-11-036). The study uses data from a survey previously conducted by the Korean Society of Critical Care Medicine for which informed consent could not be obtained due to the nature of the data collection process. Exemption and waiver of consent were deemed appropriate because the data were anonymized and used for research purposes only.

## Questionnaire

Burnout was assessed using the Maslach Burnout Inventory-Human Services Survey (MBI-HSS) [2], a globally validated instrument consisting of 22 items divided into three domains: EE (9 items), DP (5 items), and PA (8 items). Each item was rated on a 7-point Likert scale ranging from "strongly disagree" to "strongly agree". EE scores were categorized as $\leq 18$ (low), 19–26 (moderate), and $\geq 27$ (high). DP scores were categorized as $\leq 5$ (low), 6–9 (moderate), and $\geq 10$ (high). PA scores were interpreted as $\geq 40$ (low), 34–39 (moderate), and $\leq 33$ (high).

The survey was administered in Korean to ensure cultural and linguistic relevance for participants, and the psychometric properties of the Korean version of the MBI-HSS have been validated in previous study [22].

## Data and outcome

We collected data through a survey that asked respondents about their demographic information, position, salary, hospital location, and work situation. Responses were compiled and organized for analysis. In this study, an "intensivist" was defined as a physician who has completed critical care training and is currently working in the ICU, and a "critical care fellow" was defined as a physician who is undergoing specialized training in critical care medicine and is actively working in the ICU. The inclusion criteria for this study were intensivists and critical care fellows who were actively working or training in ICUs at the time of the survey. Physicians who were not actively working in ICUs during the survey period were excluded from participation. To explore potential factors associated with burnout, a comprehensive survey of demographic, professional, and institutional variables was conducted. Demographic factors included age, gender, marital status, number of children, highest level of education, religious affiliation, smoking habits, and average hours of sleep per night. Professional variables included years of critical care experience, years of service at current hospital, hours worked per week, number of night shifts per month, patient load per day, hours devoted to education, research, and training, and call frequency and annual vacation days. Institutional factors included hospital type (public vs. private), geographic location, ICU bed capacity, average patient turnover, length of stay, and mortality rates. In addition, questions examined ICU practices such as multidisciplinary rounds, the role of intensivists and fellows, and the extent of rapid response teams (RRTs).

In this study, respondents' burnout levels were evaluated using the MBI framework. In this context, positive burnout was identified as an increase in EE or DP scores or a decrease in PA scores. Severe burnout was characterized by an increase in the EE and DP scores or a decrease in the PA score.

## Statistical analyses

Categorical variables were reported as counts (n) and percentages, and continuous variables were reported as means plus standard deviations. Logistic regression analysis was used to examine the association between identified factors and the presence of burnout or high levels of burnout. Initially, univariate analysis was performed to identify factors associated with burnout, and those with a p-value of less than 0.05 were selected for inclusion in the multivariate analysis. Risk factors for burnout identified through this process were presented as odds ratios (OR) with 95% confidence intervals (CI). Statistical significance in all analyses was defined as a p-value of less than 0.05. All statistical procedures were performed using SPSS version 25.0. (IBM Corporation, Somers, NY, USA).

## Results

### Characteristics of intensive care units

The survey was conducted in the first quarter of 2019, targeting hospitals that offer subspecialty training programs in critical care medicine nationwide. A total of 51 hospitals and 79 ICUs participated in the survey. Responses were received from 194 intensivists and 59 critical care fellows actively working in these ICUs, for an overall response rate of 50.4%.

Table 1 shows the characteristics of the hospitals and ICUs as reported by the participants. Most respondents worked in adult ICUs (95.7%), with 61.7% working in medical ICUs.

**Table 1. Characteristics of intensive care units.**

| Variable | N = 253 |
|---|---|
| Location of hospital | |
| Seoul | 98 (38.7) |
| Gyeonggi/Incheon | 48 (19.0) |
| Gangwon | 11 (4.3) |
| Chungcheong | 20 (7.9) |
| Jeolla | 20 (7.9) |
| Gyeongsang | 53 (20.9) |
| Jeju | 3 (1.2) |
| Type of hospital | |
| Tertiary (public) | 60 (23.7) |
| Tertiary (private) | 124 (49.0) |
| Non-tertiary (public) | 13 (5.1) |
| Non-tertiary (private) | 56 (22.1) |
| Number of hospital beds | |
| 500 or less | 16 (6.3) |
| 501–1000 | 110 (43.5) |
| 1001 or more | 127 (50.2) |
| Type of ICU[a] by patient condition | |
| Medical | 156 (61.7) |
| Surgical | 89 (35.2) |
| Mixed | 8 (3.2) |
| Type of ICU by patient age | |
| Adult | 242 (95.7) |
| Pediatric | 11 (4.3) |

Data are presented as number of respondents and corresponding percentage of total sample (N = 253).
[a]ICU, intensive care unit.

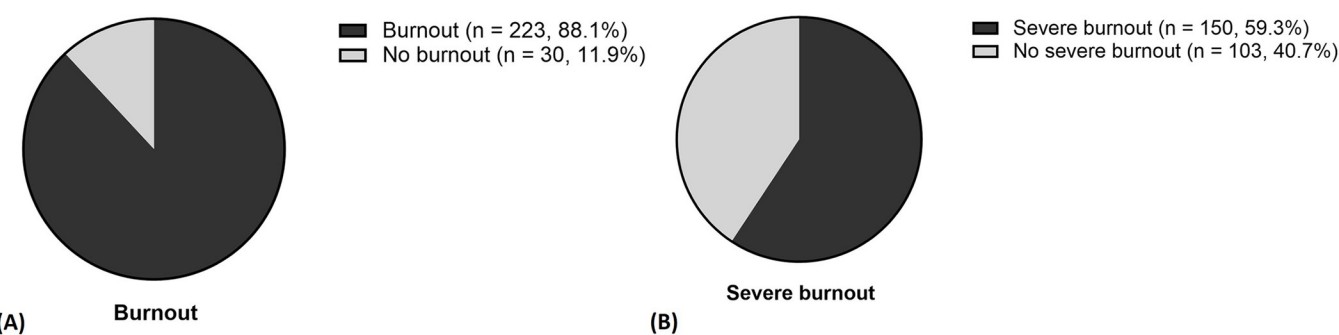

**Fig 1. Prevalence and severity of burnout among health care professionals.** Percentage and number of (A) burnout, (B) severe burnout.

## Prevalence of burnout

A high prevalence of burnout was observed among intensive care physicians in South Korea. The overall prevalence of burnout was 88.1% (223 of 253 participants), while the prevalence of severe burnout was 59.3% (150 of 253 participants), as shown in Fig 1A and 1B. S1A–S1C Fig further shows the distribution of ICU physicians across the three dimensions of burnout. Specifically, a high level of EE, DP, and reduced PA was reported by 66.0% (167 of 253 participants), 65.2% (165 of 253 participants), and 59.7% (151 of 253 participants) of the participants, respectively, indicating a generally high incidence across all dimensions of burnout.

## Demographic and professional characteristics of participants

Table 2 shows the demographic and occupational characteristics of the participants. The mean age was 41.2 years, the mean age of those with burnout was 40.8 years (p = 0.048), and the mean age of those with severe burnout was 40.4 years (p = 0.069). The sex distribution of participants was predominantly male (64.8%), and the difference in the prevalence of burnout by sex was statistically insignificant. Rank ranged from professor to fellow, with the assistant professor group being the largest at 39.9%, and a statistically significant difference was observed among those who reported experiencing burnout (p = 0.011). Most participants were married (81.0%), and the highest percentage reported being religious (66.0%). Educational qualifications varied, with a significant number of participants having master's (42.3%) and philosophy degrees (46.2%). The annual salary was divided into four categories. Moreover, the incidence of burnout between the groups differed insignificantly. Regarding length of service, most participants had between 1 and 5 years of service (56.9%), and those who experienced burnout and severe burnout were primarily in this length of service range. Length of service and burnout (p = 0.049), rather than severe burnout (p = 0.099), were significantly associated.

## Factors associated with burnout and severe burnout

Logistic regression analysis was performed to identify factors associated with burnout and severe burnout (Tables 3 and 4).

In the univariate analysis, age, position of the intensivist, employment in a medical ICU, years of experience in intensive care, frequency of stay-home night calls per month, and conflicts with colleagues in the past month were associated with burnout. Multivariate analysis further identified key factors significantly associated with burnout. The position of the intensivist, especially for those at the level of assistant professor or fellow, was associated with a higher likelihood of burnout (OR, 3.916; 95% CI, 1.485–10.327; p = 0.006). Working in a medical ICU was another strong predictor of burnout (OR, 4.557; 95% CI, 1.745–11.900; p = 0.002).

**Table 2. Demographic and professional characteristics of participants.**

| Characteristics | Total | Burnout | P-value | Severe burnout | P-value |
|---|---|---|---|---|---|
| N | 253 | 223 | | 150 | |
| Age | 41.2 ± 7.6 | 40.8 ± 7.5 | 0.048 | 40.4 ± 6.5 | 0.069 |
| Sex | | | 0.070 | | 0.636 |
| Male | 164 (64.8) | 149 (66.8) | | 99 (66.0) | |
| Female | 89 (35.2) | 74 (33.2) | | 51 (34.0) | |
| Position of intensivist | | | 0.011 | | 0.102 |
| Professor | 59 (23.3) | 47 (21.1) | | 28 (18.7) | |
| Associate professor | 34 (13.4) | 28 (12.6) | | 18 (12.0) | |
| Assistant professor | 101 (39.9) | 90 (40.4) | | 65 (43.3) | |
| Fellow | 59 (23.3) | 58 (26.0) | | 39 (26.0) | |
| Marital status | | | 0.182 | | 0.860 |
| Single, separated or widowed | 48 (19.0) | 45 (20.2) | | 29 (19.3) | |
| Married | 205 (81.0) | 178 (79.8) | | 121 (80.7) | |
| Religion | 167 (66.0) | 149 (66.8) | 0.459 | 107 (71.3) | 0.031 |
| Highest educational qualification | | | 0.069 | | 0.258 |
| Bachelor's degree | 29 (11.5) | 29 (13.0) | | 19 (12.7) | |
| Master's degree | 107 (42.3) | 95 (42.6) | | 68 (45.3) | |
| Philosophy degree | 117 (46.2) | 99 (44.4) | | 63 (42.0) | |
| Monthly salary in USD | | | 0.307 | | 0.435 |
| $3,083 –$4,624 | 57 (22.5) | 53 (23.8) | | 30 (20.0) | |
| $4,625 –$6,166 | 77 (30.4) | 69 (30.9) | | 51 (34.0) | |
| $6,167 –$7,707 | 77 (30.4) | 67 (30.0) | | 44 (29.3) | |
| Over $7,708 | 42 (16.6) | 34 (15.2) | | 25 (16.7) | |
| Length of service | | | 0.049 | | 0.099 |
| 1–5 years | 144 (56.9) | 132 (59.2) | | 88 (58.7) | |
| 6–10 years | 63 (24.9) | 55 (24.7) | | 41 (27.3) | |
| > 10 years | 46 (18.2) | 36 (16.1) | | 21 (14.0) | |

Data are presented as mean ± standard deviation or number (%), unless otherwise indicated.

Additionally, the frequency of stay-home night call per month was positively correlated with burnout (OR, 1.070; 95% CI, 1.005–1.139; p = 0.034). Conflicts with colleagues within the previous month were also significantly associated with increased burnout rates (OR, 5.344; 95% CI, 1.140–25.051; p = 0.033).

For severe burnout, multivariate analysis revealed similar factors. Among participants, those at the assistant professor or fellow level were associated with a higher likelihood of burnout (OR, 2.279; 95% CI, 1.223–4.247; p = 0.010). Frequent stay-home night calls per month remained a significant factor (OR, 1.038; 95% CI, 1.005–1.072; p = 0.022). Conflicts with colleagues in the previous month were also strongly associated with severe burnout (OR, 3.737; 95% CI, 1.693–8.250; p = 0.001). These results suggest that professional role, workload, and interpersonal relationships at work are critical determinants of severe burnout among ICU physicians.

## Discussion

This study reveals an alarmingly high prevalence of burnout among intensivists and critical care fellows in South Korea, with overall and severe burnout rates of 88.1% and 59.3%, respectively. The core components of burnout-emotional exhaustion, depersonalization, and

**Table 3. Factors associated with burnout.**

|  | Univariate analysis | | | Multivariate analysis | | |
| --- | --- | --- | --- | --- | --- | --- |
|  | OR[a] | 95% CI[b] | P-value | OR | 95% CI | P-value |
| Age | 0.957 | 0.915–1.000 | 0.052 | 1.062 | 0.968–1.166 | 0.205 |
| Male | 2.014 | 0.934–4.340 | 0.074 |  |  |  |
| Position of intensivist |  |  |  |  |  |  |
| Professor/ Associate professor | - reference- |  |  |  |  |  |
| Assistant professor/Fellow | 2.960 | 1.355–6.467 | 0.006 | 3.916 | 1.485–10.327 | 0.006 |
| Religion | 1.342 | 0.614–2.934 | 0.460 |  |  |  |
| Marital status |  |  |  |  |  |  |
| Married | 0.440 | 0.128–1.514 | 0.193 |  |  |  |
| Monthly salary in USD |  |  |  |  |  |  |
| $3,303 –$4,954 | -reference- |  |  |  |  |  |
| $4,955 –$6,606 | 0.651 | 0.186–2.278 | 0.502 |  |  |  |
| $6,607 –$8,257 | 0.506 | 0.150–1.703 | 0.271 |  |  |  |
| Over $8,258 | 0.321 | 0.090–1.148 | 0.081 |  |  |  |
| Location of hospital |  |  |  |  |  |  |
| Capital (Seoul, Gyeonggi/Incheon) | 1.658 | 0.771–3.564 | 0.196 |  |  |  |
| Type of hospital |  |  |  |  |  |  |
| Tertiary (vs. non-tertiary) | 0.790 | 0.323–1.935 | 0.606 |  |  |  |
| Type of ICU[c] |  |  |  |  |  |  |
| Medical ICU | 3.211 | 1.454–7.089 | 0.004 | 4.557 | 1.745–11.900 | 0.002 |
| Adult ICU | 1.698 | 0.349–8.262 | 0.512 |  |  |  |
| Workplace characteristics |  |  |  |  |  |  |
| Hours of sleep per day | 1.349 | 0.842–2.159 |  |  |  |  |
| Hours per week engaged in exercise | 0.909 | 0.776–1.064 | 0.233 |  |  |  |
| Hours per week engaged in teaching | 1.039 | 0.926–1.165 | 0.517 |  |  |  |
| Hours per week engaged in research | 0.983 | 0.962–1.005 | 0.139 |  |  |  |
| Member of an ICU research group | 1.023 | 0.470–2.227 | 0.955 |  |  |  |
| Vacation days taken per year | 0.933 | 0.861–1.010 | 0.088 |  |  |  |
| Years of intensive care working experience | 0.931 | 0.882–0.982 | 0.009 | 0.906 | 0.798–1.029 | 0.129 |
| Percentage of work time spent in ICU | 1.002 | 0.989–1.016 | 0.711 |  |  |  |
| Stay-in night calls per month | 0.916 | 0.788–1.066 | 0.257 |  |  |  |
| Stay-home night calls per month | 1.065 | 1.005–1.129 | 0.034 | 1.070 | 1.005–1.139 | 0.034 |
| Number of ICU patients cared for per day | 0.998 | 0.940–1.059 | 0.952 |  |  |  |
| Conflicts with colleagues in past one month | 4.807 | 1.110–20.819 | 0.036 | 5.344 | 1.140–25.051 | 0.033 |

[a]OR, odds ratio

[b]CI, confidence interval

[c]ICU, intensive care unit.

reduced personal accomplishment-have significant implications for physician well-being, patient care, and the healthcare system. Key factors associated with burnout include job position, frequency of stay-home night calls, working in a medical ICU, and recent conflicts with colleagues. These findings highlight the need for targeted interventions that address both individual and systemic factors to mitigate burnout and support critical care physicians.

The results of this study suggest that the prevalence of burnout among critical care professionals in South Korea is significantly higher than reported in many other regions. Among the intensivists and critical care fellows surveyed, 88.1% exhibited symptoms of burnout and

**Table 4. Factors associated with severe burnout.**

| | Univariate analysis | | | Multivariate analysis | | |
|---|---|---|---|---|---|---|
| | OR[a] | 95% CI[b] | P-value | OR | 95% CI | P-value |
| Age | 0.968 | 0.936–1.001 | 0.057 | | | |
| Male | 1.135 | 0.672–1.916 | 0.636 | | | |
| Position of intensivist | | | | | | |
| Professor/ Associate professor | -reference- | | | | | |
| Assistant professor/Fellow | 1.898 | 1.128–3.193 | 0.016 | 2.279 | 1.223–4.247 | 0.010 |
| Religion | 1.783 | 1.052–3.023 | 0.032 | 0.652 | 0.338–1.256 | 0.201 |
| Marital status | | | | | | |
| Married | 0.944 | 0.497–1.793 | 0.860 | | | |
| Monthly salary in USD | | | | | | |
| $3,303 - $4,954 | -reference- | | | | | |
| $4,955 - $6,606 | 1.765 | 0.875–3.564 | 0.113 | | | |
| $6,607 - $8,257 | 1.200 | 0.603–2.389 | 0.604 | | | |
| Over $8,258 | 1.324 | 0.591–2.964 | 0.496 | | | |
| Location of hospital | | | | | | |
| Capital (Seoul, Gyeonggi/Incheon) | 1.030 | 0.620–1.711 | 0.910 | | | |
| Type of hospital | | | | | | |
| Tertiary (vs. non-tertiary) | 0.709 | 0.399–1.260 | 0.241 | | | |
| Type of ICU[c] | | | | | | |
| Medical ICU | 1.110 | 0.663–1.858 | 0.691 | | | |
| Adult ICU | 1.794 | 0.533–6.042 | 0.346 | | | |
| Workplace characteristics | | | | | | |
| Hours of sleep per day | 1.055 | 0.787–1.413 | 0.720 | | | |
| Hours per week engaged in exercise | 1.030 | 0.912–1.164 | 0.634 | | | |
| Hours per week engaged in teaching | 1.034 | 0.965–1.109 | 0.344 | | | |
| Hours per week engaged in research | 0.999 | 0.980–1.018 | 0.885 | | | |
| Member of an ICU research group | 1.169 | 0.701–1.951 | 0.549 | | | |
| Vacation days taken per year | 0.947 | 0.895–1.003 | 0.063 | | | |
| Years of intensive care working experience | 0.953 | 0.912–0.996 | 0.031 | | | |
| Percentage of work time spent in ICU | 0.998 | 0.989–1.007 | 0.689 | | | |
| Stay-in night calls per month | 0.948 | 0.845–1.065 | 0.369 | | | |
| Stay-home night calls per month | 1.035 | 1.004–1.068 | 0.029 | 1.038 | 1.005–1.072 | 0.022 |
| Number of ICU patients cared for per day | 1.026 | 0.985–1.069 | 0.214 | | | |
| Conflicts with colleagues in past one month | 3.460 | 1.729–6.925 | <0.001 | 3.737 | 1.693–8.250 | 0.001 |

[a]OR, odds ratio

[b]CI, confidence interval

[c]ICU, intensive care unit.

59.3% experienced severe burnout. Specifically, high levels of EE, DP, and PA were observed in 66.0%, 65.2%, and 59.7% of respondents, respectively, highlighting the widespread impact of burnout across all dimensions. These rates exceed those reported in other countries. For example, burnout rates among intensivists in Western countries are estimated to be over 40% [11]. In France, 46.5% of ICU physicians were found to have high levels of burnout [23], while another study reported that 32% of intensivists met criteria for burnout [24]. In addition, a survey conducted in Asian ICUs found a burnout prevalence of 51% [13]. The prevalence of burnout varies widely from region to region, influenced by geographic, cultural, and

institutional factors. In this study, South Korea had a particularly high rate of burnout among critical care professionals compared to other countries. This highlights the urgent need to identify the factors that cause burnout in South Korea and to develop tailored interventions to address them.

Previous research shows that burnout is influenced by a complex interplay of personal and institutional factors. A survey of American pulmonary and critical care fellows found that half of the respondents screened positive for burnout or depressive symptoms [24]. Although burnout and depression are relatively similar, it is important to recognize them as distinct entities [25, 26]. Burnout is typically characterized by work-related disengagement and a lack of accomplishment, whereas depression involves a broader range of emotional and physical symptoms affecting various aspects of life. The high-stress environment of the ICU makes healthcare workers particularly vulnerable to these conditions. The demanding pace, emotional intensity, and critical nature of the work can exacerbate feelings of hopelessness and inadequacy, potentially culminating in depression. Particularly, a study on physicians in ICUs found that 23.8% exhibited signs of depression, with 58% of them expressing a desire to leave their jobs. Similarly, a study of critical care nurses found a correlation between scores on depression-anxiety-stress scales and emotional exhaustion, which further impacts the quality of work life [27]. Although this study does not examine depression in-depth, existing literature [28, 29] suggests that burnout factors, such as diminished PA and EE, may be closely related to depressive symptoms.

In this study, burnout among intensivists and critical care fellows was significantly associated with factors such as job position (assistant professor/fellow), working in a medical ICU, frequency of night calls per month, and recent conflicts with colleagues. These findings are consistent with previous research and highlight the multifaceted nature of burnout. Excessive workload and the burden of electronic health records have been shown to contribute to both burnout and depressive symptoms [24]. In addition, studies consistently show that working long hours increases stress and decreases job satisfaction and engagement, underscoring the need for healthcare systems to reevaluate work schedules and reduce excessive workloads [30–33]. Our findings also suggest that younger individuals in lower-level positions, such as fellows and assistant professors, may be more vulnerable to burnout. This is consistent with research indicating that younger age, being male, not having children, and being in a junior role are associated with a higher risk of burnout among ICU nurses [9, 10, 23]. A study in French adult ICUs also found that interns and residents were more likely to experience burnout than fellows or attending physicians [23]. These findings highlight the importance of targeted interventions to support younger professionals in critical care. Conflict with colleagues was also identified as a significant factor contributing to burnout. This is consistent with existing literature showing that interpersonal conflict and lack of teamwork increase stress levels, decrease job satisfaction, and increase the likelihood of burnout [1, 34–37]. Creating a collaborative and supportive work environment is critical to mitigating these risks. In addition to burnout, other phenomena such as moral injury [38] and workplace violence [39] can exacerbate stress and emotional exhaustion. Moral injury, which results from situations that violate an individual's moral values, is strongly associated with anxiety, depression, and reduced quality of life. Similarly, workplace violence, including verbal and physical abuse, has been shown to undermine trust in management, increase anxiety, and amplify feelings of anger and distress. Addressing these interrelated factors requires implementing preventive measures, fostering ethical and respectful workplaces, and establishing reporting mechanisms to effectively manage incidents. In addition, several other factors have been associated with burnout among ICU physicians, including personal characteristics such as marital status [23], number of children [23], years of ICU experience [13], and work-related factors such as working more night shifts [40], working

hours per month [13, 41], night calls [42], patient end-of-life issues [23, 40], and the coronavirus disease 2019 pandemic [43, 44]. Addressing these multiple and complex factors is essential to reducing burnout, improving physician well-being, and enhancing job satisfaction in critical care.

This study does not identify specific factors that decrease the risk of burnout; however, findings from other studies highlight several protective elements. For example, a study of pulmonary and critical care medicine fellows [24] showed that a structured coverage system (adjusted OR, 0.44; 95% CI, 0.26–0.73) and access to mental health services (adjusted OR, 0.14; 95% CI, 0.04–0.47) significantly reduced burnout. Additionally, research focusing on critical care physicians [13] found that having a religious background or beliefs, more years of service in the current department, and shift work were associated with lower burnout rates. Similarly, a study of physicians in internal medicine programs [42] found that experiencing high levels of respect from colleagues significantly reduced burnout. In response to these challenges, there are a variety of efforts to mitigate burnout. These include improving the work environment, implementing team-level interventions and stress management training programs, establishing mentoring and peer support networks, and educating healthcare workers about health and self-care [34, 45–47]. There is also a movement to optimize work schedules and limit excessive work hours to improve work-life balance [30, 48]. Efforts to integrate mindfulness and stress-reduction methods into the daily routines of healthcare workers [49, 50] are also underway. These strategies, along with organizational and cultural changes that recognize the risk of burnout among individuals, are critical to creating work environments that reduce the incidence and impact of burnout.

The impact of burnout goes beyond the well-being of healthcare professionals and can directly impact patient care and safety. In the critical care setting, burnout can impair necessary decision-making and empathy, increasing the likelihood of medical errors and potentially leading to poor patient outcomes [51, 52]. Burnout has also been linked to sleep disturbances and the willingness of healthcare workers to leave the ICU [52, 53]. Reducing burnout is, therefore, not only a matter of staff well-being but also a critical issue directly related to the healthy running of the ICU [50]. Sustainable healthcare requires ongoing efforts to reduce burnout to ensure both the well-being of healthcare workers and the quality of patient care.

Our study has several limitations. First, although our findings provide valuable insights, they may not fully represent all ICUs in South Korea, and our results may be subject to response bias. The cross-sectional nature of this study limits the ability to draw causal inferences. In addition, despite an adequate participation rate, there is a potential for selection bias, as individuals experiencing burnout or depressive symptoms may have been more likely to respond, potentially inflating the prevalence rates. The use of the MBI, while globally accepted, may also be interpreted differently in different cultural and institutional contexts, which could influence the results. In addition, this study was conducted using data collected before the COVID-19 pandemic. Other study [54] have shown that the pandemic significantly increased burnout rates, particularly among ICU nurses and female clinicians. For example, research [55] conducted during the pandemic in Korea reported that 77.3% of healthcare workers responding to the COVID-19 outbreak met criteria for burnout, with higher rates observed among females, younger individuals, and those with chronic fatigue or physical symptoms due to excessive workload. Because our study does not capture the additional burden of the pandemic, future research is needed to assess the long-term impact of COVID-19 on burnout among critical care professionals in South Korea. Despite these limitations, our findings contribute to a deeper understanding of burnout among critical care professionals in South Korea and highlight the need for targeted interventions and further research to address this growing problem.

## Conclusions

The high prevalence of burnout among intensivists and critical care fellows in South Korea highlights the urgent need for targeted interventions. The negative effects of burnout extend beyond the well-being of health care workers to affect patient safety and the efficiency of the health care system. Therefore, it is necessary to implement strategies to reduce burnout, including optimizing work schedules, strengthening support networks, and integrating stress management resources. Addressing these issues is not only essential to the well-being of health care workers but is closely linked to the delivery of quality patient care and the sustainability of the health care system. Therefore, a concerted effort must be made to reduce burnout, which is essential to maintaining the integrity of critical care services.

## Supporting information

**S1 Fig. Prevalence and severity of burnout among health care professionals.** Percentage and number of (A) emotional exhaustion, (B) depersonalization, (C) personal accomplishment. (DOCX)

## Acknowledgments

We thank all the physicians from different provinces in the Republic of Korea who participated in this survey.

## Author Contributions

**Conceptualization:** Song I. Lee, Won-Young Kim, Duk ki Kim, Gee Young Suh, Jeongmin Kim, Ha Yeon Kim, Nak-Joon Choi, Won Kyoung Jhang, Sang-Hyun Kwak, Sang-Bum Hong.

**Data curation:** Song I. Lee, Won-Young Kim, Duk ki Kim, Gee Young Suh, Jeongmin Kim, Ha Yeon Kim, Nak-Joon Choi, Won Kyoung Jhang, Sang-Hyun Kwak, Sang-Bum Hong.

**Formal analysis:** Song I. Lee, Won-Young Kim, Duk ki Kim, Gee Young Suh, Jeongmin Kim, Ha Yeon Kim, Nak-Joon Choi, Won Kyoung Jhang, Sang-Hyun Kwak, Sang-Bum Hong.

**Funding acquisition:** Song I. Lee.

**Investigation:** Jeongmin Kim.

**Methodology:** Song I. Lee, Won-Young Kim, Duk ki Kim, Ha Yeon Kim, Nak-Joon Choi, Won Kyoung Jhang, Sang-Hyun Kwak, Sang-Bum Hong.

**Visualization:** Song I. Lee, Sang-Bum Hong.

**Writing – original draft:** Song I. Lee, Won-Young Kim, Duk ki Kim, Gee Young Suh, Jeongmin Kim, Ha Yeon Kim, Nak-Joon Choi, Won Kyoung Jhang, Sang-Hyun Kwak, Sang-Bum Hong.

**Writing – review & editing:** Song I. Lee, Won-Young Kim, Duk ki Kim, Gee Young Suh, Jeongmin Kim, Ha Yeon Kim, Nak-Joon Choi, Won Kyoung Jhang, Sang-Hyun Kwak, Sang-Bum Hong.

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
