## [Decision Letter · Decision Letter 0]

1 Nov 2024

PONE-D-24-43143Burnout among Intensivists and Critical Care Fellows in South Korea: Current Status and Associated FactorsPLOS ONE

Dear Dr. Hong,

Thank you for submitting your manuscript to PLOS ONE. After careful consideration, we feel that it has merit but does not fully meet PLOS ONE’s publication criteria as it currently stands. Therefore, we invite you to submit a revised version of the manuscript that addresses the points raised during the review process.

We look forward to receiving your revised manuscript.

Kind regards,

Chinh Quoc Luong, MD., PhD.

Academic Editor

PLOS ONE

Journal Requirements:

3. Thank you for stating the following financial disclosure: “This research was supported by the Korean Society of Critical Care Medicine (Grant No. KSCCM-2024-01).”

4. We note that you have indicated that there are restrictions to data sharing for this study. PLOS only allows data to be available upon request if there are legal or ethical restrictions on sharing data publicly. For more information on unacceptable data access restrictions, please see http://journals.plos.org/plosone/s/data-availability#loc-unacceptable-data-access-restrictions. Before we proceed with your manuscript, please address the following prompts: a) If there are ethical or legal restrictions on sharing a de-identified data set, please explain them in detail (e.g., data contain potentially identifying or sensitive patient information, data are owned by a third-party organization, etc.) and who has imposed them (e.g., a Research Ethics Committee or Institutional Review Board, etc.). Please also provide contact information for a data access committee, ethics committee, or other institutional body to which data requests may be sent. b) If there are no restrictions, please upload the minimal anonymized data set necessary to replicate your study findings to a stable, public repository and provide us with the relevant URLs, DOIs, or accession numbers. For a list of recommended repositories, please see https://journals.plos.org/plosone/s/recommended-repositories. You also have the option of uploading the data as Supporting Information files, but we would recommend depositing data directly to a data repository if possible. We will update your Data Availability statement on your behalf to reflect the information you provide.

Additional Editor Comments:

Thank you for submitting your manuscript to PLOS ONE. I have completed my evaluation, and the Reviewers recommended that your manuscript be reconsidered after making major revisions. I invite you to resubmit your manuscript after addressing the comments provided below.

Reviewers' comments:

Reviewer's Responses to Questions

**Comments to the Author**

1. Is the manuscript technically sound, and do the data support the conclusions?

Reviewer #1: Yes

Reviewer #2: Partly

2. Has the statistical analysis been performed appropriately and rigorously? 

Reviewer #1: Yes

Reviewer #2: Yes

3. Have the authors made all data underlying the findings in their manuscript fully available?

Reviewer #1: Yes

Reviewer #2: No

4. Is the manuscript presented in an intelligible fashion and written in standard English?

Reviewer #1: Yes

Reviewer #2: Yes

5. Review Comments to the Author

Reviewer #1: Burnout among Intensivists and Critical Care Fellows in South Korea: Current Status

and Associated Factors

Thank you for the opportunity to review a study. It is a topic widely studied in the nursing panorama but always offers insights into the phenomenon's evolution. My comments are below:

Introduction

The background seems well set, but we should expand the epidemiological part of the phenomenon to strengthen our reasons. For example, how widespread is burnout in nurses? And in critical care nurses? For this reason, I suggest reading this paper (DOI: 10.7429/pi.2022.751028), in which stratification by item is made on a national population. Furthermore, it is unclear what the authors mean by "Intensivists" and "Critical Care Fellows". I think these figures are present in the system where the data were collected, but how do we contextualize them internationally?

The passage on cultural aspects is interesting. It is often underestimated (in the text - For example, values such as collectivism, perseverance, and hard work that are prevalent in many Asian cultures may have dual effects on burnout, potentially mitigating and exacerbating its development [13, 14].)

The study aims to examine the associated factors. It is unclear whether these factors cause burnout or the intensive environment. Clarify this point.

Broadens the meaning of the study, why is it important?

Methods

I would write a separate paragraph titled "Ethical Considerations."

I need more information about data collection. Have response rates been calculated? Emailing the link often does not let us know if the person responding is someone we want to address. Be more explicit about data collection and write a separate paragraph.

Detailed information on the instrument used should be available. Likert scale? How many items per dimension? Psychometric properties? Is the instrument available in the language of the country where it was administered? I recommend using a checklist for reporting, e.g. STROBE in the methods.

Inclusion and exclusion criteria?

Describe the population and the risk factors more accurately. Why do we call them associated factors?

Results

The results are well presented; in Table 2, try inserting the p-value in the column for each factor.

Here, you define "Risk factors for burnout and severe burnout" as risk factors. Are you congruent throughout the manuscript regarding risk factors or associated factors?

Discussion

The discussions begin with the resumption of the objective and the declaration of the key points

the discussion offers a good overview of the phenomenon, but are we sure it is burnout? In the literature, a recent review explains the relationship between different phenomena, such as moral injury; I recommend reading it (DOI: 10.1177/09697330241281376). Furthermore, are there other phenomena that interact with burnout and abandonment? For example, regarding workplace violence, I suggest reading this interesting study (DOI: 10.7429/pi.2020.732089). What are the implications for the profession? For nursing managers? Broaden the discussion on preventive interventions.

Offer a unique perspective of this article to the magazine's readers.

Reviewer #2: I would like to thank the authors for submitting a manuscript titled "Burnout among Intensivists and Critical Care Fellows in South Korea: Current Status and Associated Factors”.

This manuscript is based on a survey by the Korean Society of Critical Care Medicine to hospitals who provide CCM fellowship. Factors associated with burn out as defined by the Maslach Burnout Index are described. Below are specific comments and suggestions for improving the clarity, and educational content of the manuscript:

I would offer the following questions and/or suggestions.

Please adjust the abstract to better reflect that this survey was done by the Korean Society of Critical Care Medicine, who the target was and what the purpose of the survey was. I cant see a response rate.

Background: You write: Internationally, burnout rates among ICU professionals are alarmingly high, with studies reporting prevalence rates ranging from 33% to 47% in Western countries [9-11]. Delete the “however” at the start of the next sentence- it does not make sense. However, a recent cross-sectional survey of 159 ICUs in 16 Asian countries and regions found that the prevalence of burnout syndrome among ICU physicians and nurses in Asia accounts for 50-52% [12].

Background: You write: However, the prevalence and characteristics of burnout among ICU staff in Asian contexts, particularly in South Korea, remain understudied. Then you state: This research gap is significant given the potential impact of cultural, economic, and systemic factors on the experience of burnout. Obviously there is a study in Asian countries. Do you mean there is no data available in Korea and that is the gap? Please specify.

Background: You write: Recognizing the high prevalence of burnout among medical professionals in various specialties, this study specifically aims to address the less explored area of burnout among those at the forefront of critical care in South Korea. However, you just previously mentioned studies looking at critical care clinicians, this is confusing. Please rewrite.

Methods: The study design is unclear. Who sent out the survey. How did you know who to sent it to? Did you sent it to individuals or to institutions? You mention previous surveys. Needs to be rewritten.

Is the study IRB exempt or not. You mention an ethics approval

NownSurvey and Google Forms need to be referenced. I could not find NownSurvey online

Results: Whats the response rate?

This is unclear: The position of the intensivist, especially for those at the level of assistant professor or fellow, was associated with a higher likelihood of burnout (OR, 3.916; 95% CI, 1.485 - 10.327; p = 0.006). All surveyed people were intensivists, right?

Please delete the double brackets [[: Another study in France found the prevalence of burnout among intensivists to be approximately 32% [[16], further highlighting the disparity between regions. A similar survey from Asian ICUs found a burnout rate of 51% within the same professional cohort [[12].

In this study 47.5 % of french intenivists had burnout: Embriaco N, Azoulay E, Barrau K, et al. High level of burnout in intensivists: prevalence and associated factors. Am J Respir Crit Care Med. 2007;175:686–92.

Studies have shown female gender is associated with higher rates of burnout- see study above or (Moll V, Meissen H, Pappas S, Xu K, Rimawi R, Buchman TG, Fisher L, Bakshi V, Zellinger M, Coopersmith CM. The Coronavirus Disease 2019 Pandemic Impacts Burnout Syndrome Differently Among Multiprofessional Critical Care Clinicians-A Longitudinal Survey Study. Crit Care Med. 2022 Mar 1;50(3):440-448.)

The discussion needs to be rewritten in a more concise manner.

How can the highest qualification be only a bachelors degree in 22.5% if all participants were physicians?

How can the Stay-home night calls per month be associated with burnout but not the in house nights?

It is unclear if this survey only went to medical ICUs? There are no surgical, neuro, CVICU, etc?

The major issue though that I see is that the survey was done almost 6 years ago. Much has happened since -COVID! I would encourage the authors to repeat the survey now, in the post COVID world. The results might be different (worse?) now.

6. PLOS authors have the option to publish the peer review history of their article (what does this mean?). If published, this will include your full peer review and any attached files.

Reviewer #1: No

Reviewer #2: No

---

## [Author Response · Author response to Decision Letter 0]

19 Dec 2024

Dear Editor 

Plos one

Burnout among Intensivists and Critical Care Fellows in South Korea: Current Status and Associated Factors [PONE-D-24-43143]

I hope this letter finds you well. I am writing to submit the revised version of our manuscript titled "Burnout among Intensivists and Critical Care Fellows in South Korea: Current Status and Associated Factors", in response to the reviewers' comments and suggestions.

We have carefully considered the feedback provided by the reviewers and have made the necessary revisions to enhance the clarity and quality of our work. A detailed point-by-point response to the reviewers' comments is included with this submission. We have addressed each concern and suggestion comprehensively, and where appropriate, have revised the manuscript to reflect these changes.

We are grateful for the valuable feedback and believe that the revisions have strengthened the manuscript significantly. We hope that the revised version meets the standards of Plos one and will be considered favorably for publication.

Thank you again for granting us the additional time we requested. It would be an honor to have our work included in your esteemed publication. If you require any further information or additional changes, please do not hesitate to contact us.

<Reviewer 1>

Thank you for the opportunity to review a study. It is a topic widely studied in the nursing panorama but always offers insights into the phenomenon's evolution. My comments are below:

Thank you for reviewing our study and providing your thoughtful comments. We appreciate your recognition of the topic's relevance to the field of nursing and its potential to provide valuable insights. We have carefully considered your feedback and made revisions accordingly.

Introduction

The background seems well set, but we should expand the epidemiological part of the phenomenon to strengthen our reasons. For example, how widespread is burnout in nurses? And in critical care nurses? For this reason, I suggest reading this paper (DOI: 10.7429/pi.2022.751028), in which stratification by item is made on a national population. Furthermore, it is unclear what the authors mean by "Intensivists" and "Critical Care Fellows". I think these figures are present in the system where the data were collected, but how do we contextualize them internationally?

Answer) Thank you very much for your valuable feedback. In response, we have added information from the referenced paper (DOI: 10.7429/pi.2022.751028) to the Background to better contextualize nurse burnout. In addition, we have clarified the definitions of "intensivists" and "critical care fellows" in the Methods section to improve understanding and ensure international relevance. 

Revised manuscript (Background)

The COVID-19 pandemic has further exacerbated burnout, with one study [12] showing that more than half of nurses experienced emotional exhaustion and 85% experienced depersonalization.

Revised manuscript (Methods)

In this study, an "intensivist" was defined as a physician who has completed critical care training and is currently working in the ICU, and a "critical care fellow" was defined as a physician who is undergoing specialized training in critical care medicine and is actively working in the ICU. The inclusion criteria for this study were intensivists and critical care fellows who were actively working or training in ICUs at the time of the survey.

The passage on cultural aspects is interesting. It is often underestimated (in the text - For example, values such as collectivism, perseverance, and hard work that are prevalent in many Asian cultures may have dual effects on burnout, potentially mitigating and exacerbating its development [13, 14].)

The study aims to examine the associated factors. It is unclear whether these factors cause burnout or the intensive environment. Clarify this point.

Broadens the meaning of the study, why is it important?

Answer) Thank you for your insightful feedback. In response, we have revised the background section to explain how cultural values such as collectivism and perseverance can both mitigate and contribute to burnout, providing a clearer example of this dual effect. We have also addressed the diversity of healthcare systems and situations in Asia, explaining the importance of regional research to better understand burnout in this context, and re-emphasizing the need for research on burnout in each region due to these differences.

Revised manuscript (Background)

Despite these regional data, there is a lack of specific research focusing on South Korea. This gap is significant given the potential influence of cultural, economic, and systemic factors unique to South Korea. For example, cultural values such as collectivism and perseverance, which are prominent in South Korea, may promote social support but also increase vulnerability due to heightened expectations and self-imposed pressures [14, 15]. In addition, the diversity of health care systems and economic conditions across Asia limits the generalizability of findings from regional studies [16-18]. With its advanced healthcare system and unique workplace dynamics, South Korea warrants focused investigation to identify specific contributors to burnout and develop tailored strategies for prevention and intervention [19, 20].

Methods

I would write a separate paragraph titled "Ethical Considerations."

Answer) Thank you for your suggestion. We have added a separate section titled "Ethical Considerations" to clearly outline the ethical aspects of the study, ensuring greater transparency and alignment with ethical standards.

Revised manuscript (Method)

Ethical considerations

I need more information about data collection. Have response rates been calculated? Emailing the link often does not let us know if the person responding is someone we want to address. Be more explicit about data collection and write a separate paragraph.

Answer) Thank you for your valuable feedback and for highlighting the need for more detailed information on data collection. We have now provided additional details about the data collection process in a separate paragraph in the Methods section.

Revised manuscript (Method)

To encourage participation, the survey was announced on the Announcements section of the official website of the Korean Society of Critical Care Medicine (https://www.ksccm.org/), and additional efforts were made to engage participants by sending email and text message invitations directly to intensivists and critical care fellows. The survey responses were collected using NownSurvey (https://www.nownsurvey.com/), a proprietary online survey platform, and Google Forms, a publicly available web-based tool for the creation and management of surveys. Of the 502 individuals invited to participate, including intensivists and critical care fellows, 253 responded, resulting in a response rate of 50.4%.

Detailed information on the instrument used should be available. Likert scale? How many items per dimension? Psychometric properties? Is the instrument available in the language of the country where it was administered? I recommend using a checklist for reporting, e.g. STROBE in the methods.

Answer) Thank you very much for your valuable feedback. In response, we have included detailed information about the survey instrument used to measure burnout. The Maslach Burnout Inventory-Human Services Survey (MBI-HSS) was used, which consists of 22 items rated on a 7-point Likert scale ranging from "strongly disagree" to "strongly agree". These items assess three dimensions of burnout: emotional exhaustion, depersonalization, and personal accomplishment. The survey was administered in Korean, the native language of the participants, to ensure cultural and linguistic appropriateness. In addition, we confirmed that the psychometric properties of the Korean version of the MBI-HSS have been validated in previous studies. To increase transparency, we also referred to the STROBE checklist for reporting.

Revised manuscript (Method)

Questionnaire

Burnout was assessed using the Maslach Burnout Inventory-Human Services Survey (MBI-HSS) [2], a globally validated instrument consisting of 22 items divided into three domains: EE (9 items), DP (5 items), and PA (8 items). Each item was rated on a 7-point Likert scale ranging from "strongly disagree" to "strongly agree". EE scores were categorized as ≤ 18 (low), 19-26 (moderate), and ≥ 27 (high). DP scores were categorized as ≤ 5 (low), 6-9 (moderate), and ≥ 10 (high). PA scores were interpreted as ≥ 40 (low), 34-39 (moderate), and ≤ 33 (high).

The survey was administered in Korean to ensure cultural and linguistic relevance for participants, and the psychometric properties of the Korean version of the MBI-HSS have been validated in previous study [22].

Inclusion and exclusion criteria?

Describe the population and the risk factors more accurately. Why do we call them associated factors?

Answer) Thank you very much for your valuable feedback. We have clarified the inclusion and exclusion criteria in the Methods section. Specifically, the inclusion criteria were intensivists and critical care fellows who were actively working or training in ICUs at the time of the survey. The exclusion criteria were physicians not working in ICUs during the survey period. In addition, we have addressed your comment about "associated factors" by providing detailed information about the survey items in the Methods section.

Revised manuscript (Method)

The inclusion criteria for this study were intensivists and critical care fellows who were actively working or training in ICUs at the time of the survey. Physicians who were not actively working in ICUs during the survey period were excluded from participation.

Results

The results are well presented; in Table 2, try inserting the p-value in the column for each factor.

Answer) Thank you for your suggestion regarding Table 2. We have revised the table to include the p-value in the column for each factor as recommended.

 Revised manuscript (Table)

Table 2. Demographic and Professional Characteristics of Participants

Characteristics Total Burnout P-value Severe burnout P-value

N 253 223 150 

Age 41.2 ± 7.6 40.8 ± 7.5 0.048 40.4 ± 6.5 0.069

Sex 0.070 0.636

 Male 164 (64.8) 149 (66.8) 99 (66.0) 

 Female 89 (35.2) 74 (33.2) 51 (34.0) 

Position of intensivist 0.011 0.102

 Professor 59 (23.3) 47 (21.1) 28 (18.7) 

 Associate professor 34 (13.4) 28 (12.6) 18 (12.0) 

 Assistant professor 101 (39.9) 90 (40.4) 65 (43.3) 

 Fellow 59 (23.3) 58 (26.0) 39 (26.0) 

Marital status 0.182 0.860

 Single, separated or widowed 48 (19.0) 45 (20.2) 29 (19.3) 

 Married 205 (81.0) 178 (79.8) 121 (80.7) 

Religion 167 (66.0) 149 (66.8) 0.459 107 (71.3) 0.031

Highest educational qualification 0.069 0.258

 Bachelor’s degree 29 (11.5) 29 (13.0) 19 (12.7) 

 Master’s degree 107 (42.3) 95 (42.6) 68 (45.3) 

 Philosophy degree 117 (46.2) 99 (44.4) 63 (42.0) 

Monthly salary in USD 0.307 0.435

 $3,083 – $4,624 57 (22.5) 53 (23.8) 30 (20.0) 

 $4,625 – $6,166 77 (30.4) 69 (30.9) 51 (34.0) 

$6,167 – $7,707 77 (30.4) 67 (30.0) 44 (29.3) 

 Over $7,708 42 (16.6) 34 (15.2) 25 (16.7) 

Length of service 0.049 0.099

 1 – 5 years 144 (56.9) 132 (59.2) 88 (58.7) 

 6 – 10 years 63 (24.9) 55 (24.7) 41 (27.3) 

 > 10 years 46 (18.2) 36 (16.1) 21 (14.0) 

Data are presented as mean ± standard deviation or number (%), unless otherwise indicated. 

Here, you define "Risk factors for burnout and severe burnout" as risk factors. Are you congruent throughout the manuscript regarding risk factors or associated factors?

Answer) Thank you for pointing out the inconsistency in the use of the terms "risk factors" and "associated factors". We have carefully revised the manuscript to consistently use "associated factors" when describing statistical associations observed in the study.

Revised manuscript (Results)

Factors associated with burnout and severe burnout

Discussion

The discussions begin with the resumption of the objective and the declaration of the key points.

Answer) Thank you for your comments. We have revised the discussion to succinctly summarize the key findings and implications of the study, ensure alignment with the objectives, and highlight the critical factors associated with burnout.

Revised manuscript (Discussion)

This study reveals an alarmingly high prevalence of burnout among intensivists and critical care fellows in South Korea, with overall and severe burnout rates of 88.1% and 59.3%, respectively. The core components of burnout-emotional exhaustion, depersonalization, and reduced personal accomplishment-have significant implications for physician well-being, patient care, and the healthcare system. Key factors associated with burnout include job position, frequency of stay-home night calls, working in a medical ICU, and recent conflicts with colleagues. These findings highlight the need for targeted interventions that address both individual and systemic factors to mitigate burnout and support critical care physicians.

The discussion offers a good overview of the phenomenon, but are we sure it is burnout? In the literature, a recent review explains the relationship between different phenomena, such as moral injury; I recommend reading it (DOI: 10.1177/09697330241281376). Furthermore, are there other phenomena that interact with burnout and abandonment? For example, regarding workplace violence, I suggest reading this interesting study (DOI: 10.7429/pi.2020.732089). What are the implications for the profession? For nursing managers? Broaden the discussion on preventive interventions.

Answer) Thank you for your valuable comments and suggestions. We have discussed the interaction between burnout, moral injury, and workplace violence, reflecting the findings of the proposed papers. We have also added related content to the Discussion section.

Revised manuscript (Discussion)

In addition to burnout, other phenomena such as moral injury [38] and workplace violence [39] can exacerbate stress and emotional exhaustion. Moral injury, which results from situations that violate an individual's moral values, is strongly associated with anxiety, depression, and reduced quality of life. Similarly, workplace violence, including verbal and physical abuse, has been shown to undermine trust in management, increase anxiety, and amplify feelings of anger and distress. Addressing these interrelated factors requires implementing preventive measures, fostering ethical and respectful workplaces, and establishing reporting mechanisms to effectively manage incidents.

Offer a unique perspective of this article to the magazine's readers.

Answer) Thank you for your thoughtful comments and suggestions. To provide a unique perspective for the journal's readers, we have highlighted the novel aspects of our study, including the high prevalence of burnout among intensivists and critical care fellows in South Korea and the identification of associated factors such as job position, conflict with colleagues, and workload. In addition, we have contextualized these findings within broader phenomena such as moral injury and workplace violence, highlighting their interrelationships and implications for health care systems. We hope that this perspective will provide readers with valuable insights and stimulate further discussion on how to address burnout in critical care settings.

<Reviewer #2> 

I would like to thank the authors for submitting a manuscript titled "Burnout among Intensivists and Critical Care Fellows in South Korea: Current Status and Associated Factors”.

This manuscript is based on a survey by the Korean Society of Critical Care Medicine to hospitals who provide CCM fellowship. Factors associated with burn out as defined by the Maslach Burnout Index are described. Below are specific comments and suggestions for improving the clarity, and educational content of the manuscript:

I would offer the following questions and/or suggestions.

Thank you for taking the time to review our manuscript and provide thoughtful comments and suggestions. We sincerely appreciate your detailed feedbac

---

## [Decision Letter · Decision Letter 1]

17 Jan 2025

Burnout among Intensivists and Critical Care Fellows in South Korea: Current Status and Associated Factors

PONE-D-24-43143R1

Dear Dr. Hong,

We’re pleased to inform you that your manuscript has been judged scientifically suitable for publication and will be formally accepted for publication once it meets all outstanding technical requirements.

Kind regards,

Chinh Quoc Luong, MD., PhD.

Academic Editor

PLOS ONE

Additional Editor Comments (optional):

Reviewers' comments:

Reviewer's Responses to Questions

**Comments to the Author**

1. If the authors have adequately addressed your comments raised in a previous round of review and you feel that this manuscript is now acceptable for publication, you may indicate that here to bypass the “Comments to the Author” section, enter your conflict of interest statement in the “Confidential to Editor” section, and submit your "Accept" recommendation.

Reviewer #1: All comments have been addressed

Reviewer #2: All comments have been addressed

2. Is the manuscript technically sound, and do the data support the conclusions?

Reviewer #1: Yes

Reviewer #2: Yes

3. Has the statistical analysis been performed appropriately and rigorously? 

Reviewer #1: Yes

Reviewer #2: Yes

4. Have the authors made all data underlying the findings in their manuscript fully available?

Reviewer #1: Yes

Reviewer #2: Yes

5. Is the manuscript presented in an intelligible fashion and written in standard English?

Reviewer #1: No

Reviewer #2: Yes

6. Review Comments to the Author

Reviewer #1: Dear EIC and author,

Thank you for the opportunity to read this paper titled "Burnout among Intensivists and Critical Care Fellows in South Korea: Current Status and Associated Factors." I have read the authors' responses, which I find interesting and make the paper more fluid. I have no other comments. Best

Reviewer #2: (No Response)

7. PLOS authors have the option to publish the peer review history of their article (what does this mean?). If published, this will include your full peer review and any attached files.

Reviewer #1: No

Reviewer #2: No

---

## [Editor Report · Acceptance letter]

24 Jan 2025

PONE-D-24-43143R1 

PLOS ONE

Dear Dr. Hong, 

I'm pleased to inform you that your manuscript has been deemed suitable for publication in PLOS ONE. Congratulations! Your manuscript is now being handed over to our production team.

Kind regards, 

on behalf of

Assoc. Prof. Chinh Quoc Luong 

Academic Editor

PLOS ONE